# Influences of Psychomotor Behaviors on Learning Swimming Styles in 6–9-Year-Old Children

**DOI:** 10.3390/children10081339

**Published:** 2023-08-03

**Authors:** Renato-Gabriel Petrea, Cristina-Elena Moraru, Ileana-Monica Popovici, Ilie-Cătălin Știrbu, Liliana-Elisabeta Radu, Marin Chirazi, Cristian-Mihail Rus, Alexandru Oprean, Oana Rusu

**Affiliations:** Faculty of Physical Education and Sport, “Alexandru Ioan Cuza” University of Iași, 507184 Iași, Romania; renato.petrea@uaic.ro (R.-G.P.); gimcristinamoraru@yahoo.com (C.-E.M.); ilenuca_popovici@yahoo.com (I.-M.P.); cstirbu@uaic.ro (I.-C.Ș.); liliradu2004@yahoo.com (L.-E.R.); chirazim@yahoo.com (M.C.); cristian.rus@uaic.ro (C.-M.R.); alexandruoprean@yahoo.com (A.O.)

**Keywords:** children, manual dexterity, body schema, static balance, buoyancy, swimming, learning process

## Abstract

The aim of this study was to identify the existence of some relationships between certain psychomotor behaviors, which we consider specific to swimming, and learning to execute the technique of some swimming styles (front crawl and backstroke). The study was carried out for 10 months and included 76 children (40 boys and 36 girls) aged between 6 and 9 years who practice recreational swimming in a city in Romania. Several tools were used: the Tapping test for manual dexterity, the Goodenough test for body schema, the Flamingo test for static balance, and the horizontal buoyancy test for body balance on the water. The results indicated better ratings on all psychomotor behaviors analyzed according to gender (in favor of girls compared to boys). The levels of all analyzed psychomotor behaviors have a direct relationship to the subjects’ age. Also, we identified moderate positive correlations for manual dexterity (*rs* = 0.63 in the front crawl style; *rs* = 0.57 in the backstroke style) and strong correlations for body schema, static balance and buoyancy, coordination with the learning of the two swimming styles (*r* or rs between 0.77 and 0.85). In conclusion, psychomotor behaviors can be predictors for learning swimming styles.

## 1. Introduction

Psychomotricity can be defined, in general, as any motor action that is under the influence of mental processes, with its two sides, motor and mental, representing a unitary whole. It is dependent on the sensory, perceptive and cognitive functions, on the reception of information (analyzers) and on the appropriate execution of the response act, which determines a personal, individualized behavior [1,2,3,4,5]. Other authors [6,7] established that psychomotricity responds to human needs according to the processes of education, re-education or therapy and can be seen as a whole system based on movement (the motor act) and mental functions, conditioned both by the interaction between individuals (children–children or children–adults) and by the connection between the growth process and the education system, with effects on proper social integration [8].

Studies on motor behavior are really important because the motor part is dominant in all the activities that an individual performs. Psychomotricity can be influenced in the first years of life, up to 11–12 years [9], and is a research topic even after this age for people with behavioral or mental disorders. In this case, depending on the severity of the neuromotor disorder, psychomotricity can be a concern for their whole life [10].

The process of learning swimming styles is achieved through psychomotor development. A wide variety of motor skills are learned through everyday tasks (walking, running, throwing, catching, etc.), while swimming involves covering distances in water, in swimming pools or in open water through one of four specific styles: front crawl, backstroke, breaststroke or butterfly. The acquisition of new motor skills performed on the field or in the water causes the reorganization of the primary motor cortex, which is positively associated with the formation of motor memory.

Identifying variables that predict swimming performance is one of the main goals of the swimming “science” community [11,12]. There are studies that show that it is possible to improve performance by manipulating anthropometric, biomechanical, hydrodynamic and energetic variables [13,14,15,16,17,18]. Some swimming programs for identifying and monitoring talent, from children to elite adult swimmers, regularly include such tests [19,20]. Studies on the identification of variables that contribute to the learning of swimming styles have not been identified in the literature.

### Conduits of Psychomotricity and Their Manifestation in Swimming

The various definitions of psychomotricity [6,20,21,22,23,24] refer to a taxonomy of its components, but also a theoretical approach from different perspectives of its manifestation (education, re-education or therapy).

The components of psychomotricity influence the optimal performance of the individual who practices sports both during training and in official competitions. In the process of learning and practicing swimming, we consider the following psychomotor behaviors to be representative in this study: body scheme, static balance and balance on the water (buoyancy), general coordination, and manual dexterity.

*Body schema* is understood as an image or mental representation of one’s own body and its differentiation from space and surrounding objects in different static or dynamic situations [25,26]. It is built little by little thanks to sensorial and kinesthetic acquisitions, which are progressively integrated into the child’s cognitive life. De Meur and Staes [27] identify stages in the emergence and development of the body schema in the chronological evolution of the individual’s age. Intentional motor actions depend on the representation of the body at the level of the central nervous system (body schema), and body schema disturbances have been presented in various studies as the main subject for motor impairments [28] and for illusions that the affected body part belongs to another person [29].

In the learning process of swimming, the body schema is considered an important goal of psychomotor development that will be of great use in this instructive-educational process. The optimal age to start learning swimming styles is five years old; at this age, the child has already discovered their own body, knows their body parts and strengthened their body-spatial orientation.

*Balance* is often used in association with terms like *stability* and *postural control*. Despite the widespread use of the term, there is no universally accepted definition of human balance [30,31,32,33,34,35].

*Postural control* is a prerequisite for maintaining multiple postures in various activities [36,37]. However, balance control has been identified as being associated with three broad categories of human activity [31,38]: (a) maintaining a specific posture (e.g., standing with feet apart and arms outstretched to the side; standing on one leg with the other flexed at the knee joint; a starting position from a block start in swimming events; etc.); (b) voluntary movement performed in daily activities (motor acts that happen between two balance positions); and (c) the reaction to an external disturbance (for example: contact with an opponent during a game of handball or soccer, slipping of a foot during a change in direction, etc.). These taxonomies encompass motor actions that lead to maintaining, achieving or restoring the center of gravity (line of gravity) in the support base [39,40].

Each sport involves specific motor skills that require specific postures and movements to be completed [41,42,43]. Balance is an important factor in many athletic skills, but the relationship between results in sports competitions and balance is not yet fully understood [44,45]. A lower level of balance is associated with injuries such as sprains and tears of muscles, tendons and ligaments [41].

*Body balance on water (buoyancy)* not only determines whether the body “sinks or floats in water”, but also affects the stability/balance of the body in the fluid. Depending on the locations of the center of gravity (CG) and center of buoyancy (CB), the body can be stable, neutral, or unstable in water [46]. Barbosa et al. [47] identified the *hydrostatic profile* of swimmers as the ability to float, and the *hydrodynamic profile* of swimmers as the ability to “glide” on water, which can be determined by buoyancy and various drag forces. Some studies suggest that artificially increasing buoyancy can improve swimming performance and that subjects who naturally have a higher amount of body fat may have an innate advantage in swimming performance [48].

In the learning process of swimming, the body balance on the water is important and manifests itself in two ways. On the transverse axis of the trunk, the loss of balance occurs laterally (left–right) (see the position of the shoulder blades in the positions of the front crawl and backstroke). On the longitudinal axis of the trunk, the loss of balance is determined by the immersion (“falling”) of the legs in the water, especially in the breaststroke and butterfly styles due to the extension of the head and trunk to ensure breathing, but also imposed by the regulation (in the case of the breaststroke).

*Coordination* includes oculo-motor and general dynamic coordination, and consists of the ability to associate movements in order to ensure efficient motor acts [9,49,50,51]. In order for movement to be adapted to a purpose, it must occur harmoniously in time and space, so it must be coordinated [52,53,54]. Eye-motor coordination is the basis on which prehension is built (the act of grasping with the fingers). Hand–eye coordination is developed and refined with positive effects on the control and improvement of gestures [6].

The *general dynamic coordination* is achieved with the help of motor skills supporting strength, speed, resistance and flexibility. Coordination of movements occurs only through constant repetition and gradually develops as the child grows. Control of coordinated activity is achieved through the feedback mechanism of proprioception and subcortical centers.

In swimming, coordination is a very important component. However, in aquatic locomotion (e.g., front crawl style), 85% of propulsion is due to the actions of the upper limbs [55], making their coordination very important. Coordination between arms is related to swimming speed and changes due to breathing, buoyancy (external forces) and balance of body segments [54,56,57]. The execution technique of the four swimming styles is based on elements of coordination of the upper limbs among themselves, of the lower limbs among themselves and of all the limbs as a whole. To these, the twisting or extension movements of the head are added to allow inspiration to be achieved.

*Manual dexterity* is a term used to explain a range of different hand and finger skills and performances [58]. These skills include reaction time, hand preference (dominant), wrist flexion speed, finger touch speed, aiming, hand stability and arm stability [59].

In the process of learning to swim, manual dexterity is not as important as it is in other sports (volleyball, handball, basketball, etc.), showing itself in the active phase of the upper limb when the hand grabs, pulls and pushes the water.

## 2. Materials and Methods

### 2.1. Aim and Hypotheses

The *purpose* of this study is to identify what type of relationship exists between psychomotor behaviors, as independent variables, and learning the execution technique of some swimming styles (front crawl and backstroke), as dependent variables.

Two main working hypotheses were formulated, each with 5 secondary hypotheses.

**Hypothesis** **H1.**
*There is a positive relationship between psychomotor behaviors and learning front crawl style.*


**Hypothesis** **H2.**
*There is a positive relationship between psychomotor behaviors and learning backstroke style.*


### 2.2. Research Subjects

The study was carried out between October 2021 and July 2022, with a population of 428 children practicing leisure swimming (beginner level) at one of the swimming pools in the city and metropolitan area of Iași, Romania. The research subjects are represented by 76 children (17.75% of the population), aged between 6 and 9.11 years (M = 7.2 years, SD = ±1.1), distributed according to gender (40 boys (52.6%), 36 girls (47.4%)) and age (19 (25%) for each category: 6.0–6.11 years, 7–7.11 years, 8–8.11 years and 9–9.11 years), who participated and completed all the tests included in the study. For each participant, the parents expressed and signed their consent (see Table 1).

The research subjects followed a program of two lessons per week, lasting 60–75 min each, following a program that included different exercises structured by learning stages for the two swimming styles, front crawl and backstroke.

The study was conducted in accordance with the Oviedo Convention of 1997 and the Helsinki Declaration of 1964, with the approval of the Ethics Committee of the Faculty (protocol code 13bis/30 March 2021). Subjects were included with prior written consent obtained from a parent or guardian for each child.

### 2.3. Measurement and Evaluation of Variables

All the tests used in the process of measuring and evaluating the variables (independent and dependent) were carried out in the same swimming center. Specific tests were used for each individual variable, which required specific materials, prior instruction, individual subject testing, measurement and evaluation (reference to rating standards). Testing was completed without runtime pressure. After the creation of the sample, at the end of the first month of research, the psychomotor tests were applied in the technical meeting space of the swimming center in the following sequence: for manual dexterity—the Tapping test, for the body schema—the Goodenough test, for static balance—the Flamingo test, for buoyancy—the horizontal buoyancy test, and for coordination—10 tests (5 for the lower limbs and 5 for the upper ones) and the Matorin test. The execution technique testing of the two swimming styles was carried out at the end of the research period in the swimming pool (25 m length, working corridor—2.5 m, and water temperature 26–28 °C), and the elements of the execution technique were measured in different phases. The assessment of the styles was carried out by two coaches, other than those involved in the training process, performing an inter-rater fidelity test. The full description of the tests is included in Appendix A.

### 2.4. Statistical Analyses

The results from this research were stored and processed using IBM SPSS 20 software (IBM Corp, Armonk, NY, USA). Descriptive statistical analyses allowed for checking the condition of normal data distribution, by calculating the Kolmogorov–Smirnov (K-S) and Shapiro–Wilk (S-W) coefficients. To check the correlation between the variables, the Pearson coefficient (*r*) for variables with parametric normal distribution and the Spearman coefficient (*rs*) for variables with non-parametric non-normal distribution were calculated. According to [60,61], the interpretation of the obtained values of *r*/*rs* was carried out as follows: between 0.01 and 0.09—negligible positive relationship; *r/rs* between 0.10 and 0.39—weak positive relationship; *r*/*rs* between 0.40 and 0.69—moderate positive relationship; *r*/*rs* between 0.70 and 0.89—strong positive relationship; and *r*/*rs* between 0.90 and 1.00—very strong positive relationship. 

When *r* has a negative value, we use the same interpretation range with the specification that the association is negative. In addition, due to the inconsistency of the correlation coefficient, the coefficient of determination (the square of the correlation coefficient *r*^2^) was also calculated to establish the proportion of influence of the independent variable on the dependent one [61].

The confidence interval taken into account was 95%.

## 3. Results

### 3.1. Descriptive Statistical Analysis

Depending on the gender variable, the results suggest that girls recorded better values for all the independent and dependent variables analyzed (see Table 2).

For the manual dexterity variable, girls scored 1.48 centiles higher than boys on the Tapping test. In the case of body schema, the data obtained from the Goodenough test show that girls obtained values 1.16 better than those obtained by boys. For the static balance variable, the difference in values obtained by girls compared to boys in the Flamingo test is 1.34 in favor of girls, while for body balance on water, it is 1.45 greater, also for girls. For the general coordination variable, girls recorded values 1.09 higher than boys.

In the case of the dependent variables, the values obtained in the tests are in favor of the girls, 1.19 for the front crawl style and 1.50 for the backstroke style.

By age variable, the results suggest that psychomotor skills are related to children’s increasing age (see Table 3). The older the child, the higher the values. Thus, we present the data below, relating them to the age category 9–9.11 years.

In the measurement of the manual dexterity variable, we found the following differences between the age categories: 3.69 compared to those aged 8–8.11 years, 9.74 compared to those aged 7–7.11 years, and 19.74 compared to those aged 6–6.11 years.

For the body schema variable, the recorded differences are 3.16 compared to those aged 8–8.11 years, 5.26 compared to those aged 7–7.11 years, and 9.11 compared to those aged 6–6.11 years.

For the static balance variable, a difference of 1.2 was recorded compared to those aged between 8 and 8.11 years, 3.17 years compared to 7–7.11 years, and 4.53 compared to those between 6 and 6.11 years. In the case of the body balance on water variable, there are differences of 1.78 compared to those aged 8–8.11 years, 3.13 compared to those aged 7–7.11 years, and 4.87 compared to those aged 6–6.11 years.

In the case of the coordination variable, a difference of 1.1 was recorded compared to those between 8 and 8.11 years, 2.58 compared to those between 7 and 7.11 years, and 4.79 compared to those between 6 and 6.11 years.

The differences between the age categories for the technique variable of the front crawl style were as follows: 1.48 compared to those between 8 and 8.11 years old, 2.95 compared to those between 7–7.11 years old, and 4.47 compared to those between 6 and 6.11 years old. For the backstroke style variable, the differences were values of 1.37 compared to those aged 8–8.11 years, 2.42 compared to those aged 7–7.11 years, and 3.53 compared to those aged between 6 and 6.11 years. 

### 3.2. Inferential Statistical Analysis-Hypothesis Testing

Checking the normality of the seven variables analyzed in the study was performed using the Kolmogorov–Smirnov and Shapiro–Wilk statistical tests. The obtained values show that three of them have a normal distribution: body schema, body balance on water and front crawl style technique. The other four have a distribution that does not meet the condition of normality: manual dexterity, static balance, general coordination and backstroke style technique (see Table 4).

Two main hypotheses were formulated in the study, each with five secondary hypotheses.

**Hypothesis** **H1.**
*There is a positive relationship between psychomotor behaviors and learning front crawl style technique.*


Hypotheses derived from hypothesis H1:

**Hypothesis** **H1a.**
*There is a positive relationship between manual dexterity and learning front crawl style technique.*


**Hypothesis** **H1b.**
*There is a positive relationship between body schema and learning front crawl style technique.*


**Hypothesis** **H1c.**
*There is a positive relationship between static balance and learning front crawl style technique.*


**Hypothesis** **H1d.**
*There is a positive relationship between body balance on the water (buoyancy) and learning front crawl style technique.*


**Hypothesis** **H1e.**
*There is a positive relationship between general coordination and learning front crawl style technique.*


The obtained statistical values of the coefficients calculated in the testing of the hypotheses derived from H1 suggest statistically significant positive relationships between all the independent variables and the dependent variable. In the case of manual dexterity, a moderate relationship was identified with the dependent variable (learning the front crawl style technique) (*rs* = 0.63, *r*^2^ = 0.40, *p* = 0.001). In the case of the other variables, the data show statistically significant positive relationships with a strong intensity: for body schema (*r* = 0.80, *r*^2^ = 0.64, *p*= 0.001), for static balance (*rs* = 0.82, *r*^2^ = 0.67, *p* = 0.001), for buoyancy (*r* = 0.78, *r*^2^ = 0.61, *p* = 0.001), and for general coordination (*rs* = 0, 81, *r*^2^ = 0.65, *p* = 0.001). Therefore, we can state that the derived hypotheses and the hypothesis H1 were confirmed (see Table 5).

**Hypothesis** **H2.**
*There is a positive relationship between psychomotor behaviors and learning backstroke style technique.*


Hypotheses derived from hypothesis H2:

**Hypothesis** **H2a.**
*There is a positive relationship between manual dexterity and learning backstroke style technique.*


**Hypothesis** **H2b.**
*There is a positive relationship between body schema and learning backstroke style technique.*


**Hypothesis** **H2c.**
*There is a positive relationship between static balance and learning backstroke style technique.*


**Hypothesis** **H2d.**
*There is a positive relationship between body balance on the water (buoyancy) and learning backstroke style technique.*


**Hypothesis** **H2e.**
*There is a positive relationship between general coordination and learning backstroke style technique.*


In the case of testing hypothesis H2, the Spearman statistical coefficient was calculated, taking into account the non-normal distribution of the results of some variables. Thus, a relationship of moderate intensity was identified between the variable manual dexterity and learning the backstroke style technique (*rs* = 0.57, *r*^2^ = 0.32, *p* = 0.001). For the other derived hypotheses (H2b, H2c, H2d, H2e), the results show the existence of positive, statistically significant, strong relationships between the dependent variable (learning the backstroke technique) and the independent variables: body schema (*rs* = 0.77, *r*^2^ = 0.59, *p* = 0.001), static balance (*rs* = 0.81, *r*^2^ = 0.65, *p* < 0.001), buoyancy (*rs* = 0.85, r^2^ = 0.72, *p* < 0.001), and general coordination (*rs* = 0.78, *r*^2^ = 0.61, *p* < 0.001). Therefore, we can state that the derived hypotheses and hypothesis H2 have been confirmed (see Table 6).

## 4. Discussion

In this study, we aimed to verify the influences of some psychomotor behaviors, as independent variables (manual dexterity, body schema, static balance, body balance on the water-buoyancy, and general coordination), in the learning process of the technical execution of front crawl and backstroke swimming styles, as dependent variables.

The results of our study for the “Tapping” test suggest that the subjects recorded a higher level of manual dexterity in girls compared to boys and in the older age group (9–9.11 years) compared to the other three age groups analyzed. These data agree with the values obtained in other studies [62,63,64,65]. The better level recorded in girls compared to boys is also confirmed by the study of Junaid and Fellowes [66] who observed that in children between 6 and 8 years, boys develop ball skills earlier than girls and that girls acquire manual dexterity before boys. In another study, Jaime et al. [67] found an association between manual dexterity and age in children between 4 and 15 years.

The values obtained for the assessment of the body schema showed a better level in girls compared to boys. These results agree with previous studies [68,69,70]. Our results show that depending on age, older subjects have better representations of their own body compared to younger ones, with the largest differences being between the 9–9.11 years and 6–6.11 years categories. The study carried out by Șunei et al. [71] measured body schema using the Draw-a-Person Test and found statistically significant differences between girls and boys aged 5 to 8 years. Thus, girls appear to have more knowledge about body dimensions (e.g., tall, short, thin, fat, etc.) or a greater ability to judge their body in terms of size and shape compared to boys [72]. León et al. [72] also highlight the interdependence between age and body schema in children aged between 4 and 15 years. Children’s judgments about the sizes of body segments and the body as a whole—the body schema—improve with age, and the period of 6–9 years is the one in which the image of one’s own body in space and time takes shape. At preschool ages, body image is unstable, and children’s cognitive limitations could explain the lack of a good body schema representation.

The results obtained in the Flamingo test, which measured the static balance variable, indicate a good level and is better in girls compared to boys. Girls have a better static balance than boys by 1.34 s, and the older the child, the better the level of balance. Our findings are consistent with the synthesis of Schedler et al. [73] who concluded that for subjects aged 6–18 years, girls perform better than boys in showing static balance, and boys perform slightly better than girls in showing dynamic balance and pro-active balance. Schedler et al. [73,74] identified that adolescents (14–18 years) have better balance performance (static, dynamic and proactive) than children (6–13 years). Other studies [75,76] claim that teenagers show better postural control than children and contradict some theories that claim that body balance is outlined around the age of ten. The maturity of human balance is not completed in childhood and may last until adolescence or young adulthood [73]. The age period between 6 and 8 years is considered a transitional phase in the development of postural control, when balance performance suddenly increases, which has been attributed to better sensory and motor manifestation, as well as changes in postural control strategies [77,78].

The values obtained in the body balance on water test suggest that girls register a better level compared to boys. McLean and Hinrichs [79] concluded, based on a study identifying the distance (d) between the center of gravity (CG) and the center of buoyancy (CF) in male and female swimmers aged 18–19 years, that girls have a much smaller distance compared to boys and, implicitly, better buoyancy. Depending on the locations of the CG and the CF, the body can be either stable, neutral or unstable [46]. The body is stable when it remains at the surface of the water, it is neutral when it balances somewhere in the water mass, and unstable when it sinks completely. And, in our study, girls have better buoyancy than boys, as shown by maintaining body balance on water for 1.45 s longer. In swimming, anthropometry influences hydrostatics and hydrodynamics, these influence the execution technique, which in turn, influences energy resources, all of which are reflected in swimming performance [11].

Although, in this study, age differences are recorded (those in the older age category compared to the other categories), we have not identified research that shows that this variable would be a predictor for body balance on water buoyancy, and therefore we can compare the results.

As with the other independent variables analyzed, the data indicate differences in the level of general coordination by gender, with 1.09 points in favor of girls. Battaglia et al. [80] concluded that there were significant differences in motor quotient scores between girls and boys. But, other studies suggest significant differences in motor skills between boys and girls, in favor of boys [81,82]. Boys perform better than girls, which could be related to both daily physical activities and playing sports [82,83].

Our data indicate a level of general coordination directly related to age. The older the children (in the age category 9–9.11 years), the better the level of coordination. The results are similar to those of other studies [82], which suggest that the level of coordination increases with age for both genders. Additionally, higher levels of coordination during childhood and adolescence influence children’s ability to successfully participate in movement situations and engage in physical activity throughout life [84,85].

Similar to the results obtained in the tests of the independent variables, the scores achieved in learning the two swimming styles, as dependent variables, are in favor of girls and children of the older age group.

Testing the two main hypotheses of the study, which examine the existence of a positive relationship between the psychomotor components and the learning of the two swimming styles, the results showed values of *r* or *rs* between 0.57 and 0.85. If a moderate positive correlation was recorded on the manual dexterity variable (*rs* = 0.63 for front crawl style; *rs* = 0.57 for backstroke style), for body schema, general coordination, static balance and body balance on water, the values are between 0.77 and 0.85 for both styles. This suggests a moderate–strong relationship between the variables.

There are studies in the literature that show that it is possible to improve sports performance by manipulating anthropometric, biomechanical and/or energetic variables [13]. Also, anthropometric, hydrostatic and hydrodynamic variables are described as being related to swimming performance [14,15,16,17]. Anthropometric, hydrodynamic and biomechanical testing procedures are often reported in the literature in attempts to predict swimming performance, as in other sport disciplines [18]. The literature is no longer as “rich” regarding psychomotor behaviors and their association with swimming (learning or improvement).

The novelty of this research comes from the fact that it proposes the association of these variables with swimming and also approaches the dependent variable as the learning process of swimming styles and not the sports performance in swimming. Information about this process is missing in the specialized literature, although it is the basis on which future sports performance is built.

In carrying out this study, a series of aspects were identified that can be considered as limitations of the research. The limitation of the research sample was due to either the choice of the parents not to express their consent for the engagement of the minors in the study, or by dropping out of the swimming lessons along the way. Also, some results could have been negatively influenced by certain environmental barriers (noises, etc.) by decreasing the subjects’ ability to concentrate.

Future research directions can be oriented towards the identification of causal relationships, through quantitative analyses, or linear or multiple linear regression, between the invoked variables. Also, other psychomotor components (spatio-temporal organization and laterality) could be taken into account, which would provide a much more detailed picture regarding their influences in the process of learning swimming styles. The research could be extended to the other two swimming styles, but also for other invoked variables (other age categories and other swimming categories—open water swimming and synchronized swimming).

## 5. Conclusions

The study highlighted gender differences, in favor of the girls, as well as age differences, in favor of the older children, both for all the psychomotor components and for learning the technical execution of the two swimming styles.

All the psychomotor behaviors invoked in the study (manual dexterity, body schema, static balance, body balance on the water-buoyancy and general coordination) had positive relationships with learning the technical execution of swimming styles, front crawl and backstroke. Manual dexterity had positive associations of moderate intensity (both front crawl and backstroke styles), and the other variables had positive associations of strong intensity with both styles. Therefore, they are very important in the process of learning swimming styles (front crawl and backstroke). The main hypotheses of the study were confirmed.

The study gave us the opportunity to analyze the balance of the body on water buoyancy from a psychomotor perspective, not just from a hydrostatic and dynamic perspective, when learning swimming procedures.

## Figures and Tables

**Table 1 children-10-01339-t001:** Research subjects.

Age	Total Subjects			
	Male		Female	
	No	%	No	%
6–6.11 years	10	52.6	9	47.4
7–7.11 years	11		8	
8–8.11 years	10		9	
9–9.11 years	9		10	
Total	40		36	
	76 (100%)			

**Table 2 children-10-01339-t002:** The results of the descriptive statistical analysis by gender.

Variables/Tests	Subjects			Male			Female		
	M	SD	ES	M	SD	ES	M	SD	ES
Manual dexterity—Tapping test	74.08	±10.09	±1.15	73.38	±10.21	±1.61	74.86	±10.03	±1.67
Body schema—Goodenough test	19.20	±3.57	±0.41	18.65	±3.59	±0.56	19.81	±3.48	±0.58
Static balance—Flamingo test	11.84	±2.08	±0.23	11.20	±2.12	±0.33	12.54	±1.82	±0.30
Body balance on water (buoyancy)—the horizontal buoyancy test	18.48	±2.10	±0.24	17.79	±1.74	±0.27	19.24	±2.23	±0.37
General coordination—10 tests + Matorin test	12.09	±2.11	±0.24	11.58	±2.38	±0.377	12.67	±1.62	±0.27
Front crawl style execution technique	20.82	±2.08	±0.24	20.25	±2.14	±0.33	21.44	±1.85	±0.31
Backstroke style execution technique	20.96	±1.69	±0.19	20.25	±1.64	±0.26	21.75	±1.39	±0.23

**Table 3 children-10-01339-t003:** The results of the descriptive statistical analysis by age.

Variables/Tests	6–6.11 Years			7–7.11 Years			8–8.11 Years			9–9.11 Years		
	M	SD	ES	M	SD	ES	M	SD	ES	M	SD	ES
Manual dexterity—Tapping test	62.63	±7.14	±1.63	72.63	±3.05	±0.70	78.68	±5.22	±1.20	82.37	±10.05	±2.30
Body schema—Goodenough test	14.47	±1.26	±0.29	18.32	±0.88	±0.20	20.42	±1.07	±0.24	23.58	±1.83	±0.42
Static balance—Flamingo test	9.53	±1.04	±0.24	10.89	±1.40	±0.32	12.86	±1.21	±0.27	14.06	±0.85	±0.19
Body balance on water (buoyancy)—the horizontal buoyancy test	16.05	±0.64	±0.14	17.79	±1.03	±.23	19.14	±1.26	±0.29	20.92	±1.38	±0.31
General coordination—10 tests + Matorin test	9.42	±1.34	±0.30	11.63	±1.06	±0.24	13.11	±1.04	±0.24	14.21	±1.03	±0.23
Front crawl style execution technique	18.37	±1.21	±0.27	20.16	±1.06	±0.24	21.63	±1.01	±0.23	23.11	±1.24	±0.28
Backstroke style execution technique	19.26	±1.19	±0.27	20.37	±1.11	±0.25	21.42	±1.01	±0.23	22.79	±1.03	±0.23

**Table 4 children-10-01339-t004:** Tests for checking the normality of data distribution.

Research Variables	Kolmogorov–Smirnov			Shapiro–Wilk			Distribution
	K-S	df	Sig.	S-W	df	Sig.	
Manual dexterity	0.146	76	0.000	0.926	76	0.000	not normal
Body schema	0.096	76	0.078	0.979	76	0.248	normal
Static balance	0.104	76	0.040	0.964	76	0.031	not normal
Body balance on the water (buoyancy)	0.092	76	0.176	0.968	76	0.054	normal
General coordination	0.127	76	0.004	0.958	76	0.013	not normal
Front crawl style technique	0.101	76	0.053	0.974	76	0.118	normal
Backstroke style technique	0.123	76	0.006	0.956	76	0.010	not normal

**Table 5 children-10-01339-t005:** The correlations between the independent variables and the technical execution of the front crawl style.

Independent Variables	Dependent Variable	Coefficient of Correlation	Coefficient of Determination	Association Level	Sig.
Manual dexterity	Technical execution of front crawl style	*rs* = 0.63	*r*^2^ = 0.40	moderate	*p* = 0.001
Body schema		*r* = 0.80	*r*^2^ = 0.64	strong	
Static balance		*rs* = 0.82	*r*^2^ = 0.67	strong	
Body balance on the water (buoyancy)		*r* = 0.78	*r*^2^ = 0.61	strong	
General coordination		*rs* = 0.81	*r*^2^ = 0.65	strong	

**Table 6 children-10-01339-t006:** The correlations between the independent variables and the technical execution of the backstroke style.

Independent Variables	Dependent Variable	Coefficient of Correlation	Coefficient of Determination	Level of Association	Sig.
Manual dexterity	Technical execution of backstroke style	*rs* = 0.57	*r*^2^ = 0.32	moderate	*p* = 0.001
Body schema		*rs* = 0.77	*r*^2^ = 0.59	strong	
Static balance		*rs* = 0.81	*r*^2^ = 0.65	strong	
Body balance on the water (buoyancy)		*rs* = 0.85	*r*^2^ = 0.72	strong	
General coordination		*rs* = 0.78	*r*^2^ = 0.61	strong	

## Data Availability

The data are available upon request from the corresponding author.

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
