# Peer review of "Influences of Psychomotor Behaviors on Learning Swimming Styles in 6–9-Year-Old Children"

_children, 2023, doi:10.3390/children10081339_

Round 1
Reviewer 1 Report (Previous Reviewer 3)
Dear authors.
Thank you for your effort to provide a revised version of your manuscript. Your manuscript has been improved. However, there is still serious issues related to the general form (much too long! It must be synthetized as recommended and requested last time). Still too much references (more than 130!!!).
Correct.
Author Response
Thank you for your time and appreciation!
Taking into account all the reviewers, we have reorganized the information in the article. Thus, the Results chapter was reorganized for a clear and concise presentation. Where we could, the results were also presented in the form of a table, and a much more concise description. The Discussion chapter has also been revised, referring to our results and comparing them with those obtained in previous studies. The conclusions were revised, in the sense of extracting the clearest possible messages of what was obtained in the study.
The number of bibliographic sources has been reduced to the maximum possible.
Given the length of the article, we considered it necessary to eliminate some information - those related to laterality and spatio-temporal organization, with the intention of detailing the statistical analysis in a future study.
Reviewer 2 Report (New Reviewer)

The English is very well written. Additional changes could be made only regarding writing style and minimal English writing, but overall, the text is very well enrolled.
Author Response
Thank you for your time and appreciation!
Taking into account all the reviewers, I reorganized the information in the article. So,
- we took into account your suggestion regarding the title of the article.
- Given the length of the article, we considered it necessary to remove some information - those related to laterality and spatio-temporal organization, with the intention of detailing the statistical analysis in a future study.
- At Research subjects, a table with the distribution of the incubated variables was included.
- In the Measurement of variables, the information regarding the tests used for each analyzed psychomotor behavior was rewritten. Thus, the tests were briefly presented, and in Appendix A all the details followed in the measurement process were included.
- At Statistical Analysis, the information regarding the tests and the calculated coefficients were synthesized, referring to the intervals identified to report the results.
- In the Results, the information was reorganized, the results being presented in a synthetic descriptive form, as well as in tables.
- The Discussion chapter was also revised, referring to our results and the comparison with those obtained in previous studies.
- The conclusions were revised, in the sense of extracting as clear messages as possible of what was obtained in the study.
- The number of bibliographic sources has been reduced to the maximum possible.
- The article was revised in the English version.
Reviewer 3 Report (New Reviewer)
Introduction
The introduction is quite lengthy and could be condensed to provide a more concise overview of the study's objectives and scope.
The article lacks a clear research question or hypothesis, making it challenging for readers to understand the specific objectives and outcomes expected from the study.
While the article references theories like the Cattell-Horn-Carroll theory, it does not clearly establish how these theories relate to the study or how they inform the research questions.
The introduction could benefit from a more focused and structured organization to enhance readability and coherence.
The references mentioned in the introduction need to be cited appropriately throughout the text to support the statements and claims made in the article.
The introduction does not provide any information about the study's methodology or data collection procedures. The absence of details on how the study was conducted raises questions about its reliability and validity
Methods
The manuscript should explicitly describe how the data were collected, including the specific psychomotor tests used, the order in which the tests were administered, and any relevant instructions given to the participants.
The manuscript should include detailed demographic information about the participants, such as age, gender, and any relevant background characteristics that could potentially influence the study outcomes.
The specific protocol tests used for measuring psychomotor behaviors and swimming style learning should be thoroughly described within the main manuscript. This includes detailed explanations of the test procedures, scoring methods, and any established rating standards used.
Results
The study mentions some non-significant findings, such as the lack of significant differences between hand laterality and foot laterality in backstroke technique learning, but it does not provide an interpretation or possible reasons for these outcomes.
The Results section would benefit from the inclusion of figures or tables to present the key findings visually.
The section repeats statistical details for each correlation tested, which can make the content cumbersome and less engaging.
Discussion
The Discussion section occasionally makes general statements without providing specific evidence or data to support them. For example, when discussing coordination differences between boys and girls, it mentions that boys perform better than girls in motor quotient scores but does not elaborate on the specific findings from their study to validate this claim.
While the authors interpret the correlations between psychomotor behaviors and swimming technique learning, the discussion remains relatively superficial.
The authors emphasize the uniqueness of their research, but they could enhance the discussion by drawing comparisons with similar studies involving other sports or physical activities.
While the authors briefly mention future research possibilities, they could expand on potential hypotheses or specific research designs that might further explore the relationships between psychomotor behaviors and swimming technique learning.
While the overall language is understandable, there are areas where sentence structure, word choices, and grammar could be improved for clarity and coherence. Some sentences may be too wordy or convoluted, and there are instances of repetitive phrasing that could be refined.
Author Response
Thank you for your time and appreciation!
Taking into account all the reviewers, I reorganized the information in the article. So,
- we took into account your suggestion regarding the title of the article.
- Given the length of the article, we considered it necessary to remove some information - those related to laterality and spatio-temporal organization, with the intention of detailing the statistical analysis in a future study.
- At Research subjects, a table with the distribution of the incubated variables was included.
- In the Measurement of variables, the information regarding the tests used for each analyzed psychomotor behavior was rewritten. Thus, the tests were briefly presented, and in Appendix A all the details followed in the measurement process were included.
- At Statistical Analysis, the information regarding the tests and the calculated coefficients were synthesized, referring to the intervals identified to report the results.
- In the Results, the information was reorganized, the results being presented in a synthetic descriptive form, as well as in tables.
- The Discussion chapter was also revised, referring to our results and the comparison with those obtained in previous studies.
- The conclusions were revised, in the sense of extracting as clear messages as possible of what was obtained in the study.
- The number of bibliographic sources has been reduced to the maximum possible.
- The article was revised in the English version.

Round 2
Reviewer 1 Report (Previous Reviewer 3)
Thank you for your revised version. All corrections made have improved the overall quality of the manuscript. However, the introduction still MUST be syntherized again (arround 3 pages, it is not acceptable for a research article). The list of references still MUST be reduced. Indeed, 99 references is still to much for a research artcile).
Finally, the title has been mofified. I suggest to replace it by a shorter one' It could be, for example, "Influences of Psychomotor Behaviors on Learning Swimming Styles in 6-9-Year-Old Children".
Better to be straight to the point!!! However, it is just my simple advice...
Dear Editor,
The manuscript has been corrected. However, it still need to be improved before considering it for publication.
Best regards,
Olivier Hue
Author Response
Thank you for taking the time and effort to review our article!
Following the new requests, we made the following changes:
- even if there are several shades (being 4 swimming styles of which only 2 have been covered in the article), we agreed with your idea to change the title to a simpler version
- the Introduction chapter underwent some changes, as well as the number of sources in the bibliographic list. Because we used measurement and evaluation tools for 5 of the psychomotor behaviors, we considered it necessary to present these concepts as briefly as possible in the introductory part.
- The topic of psychomotor behaviors takes up an important space in the literature. Therefore, it required a considerable synthesis effort. Hence, the large number of sources included in the list of references. We excluded everything we considered unnecessary.
- the reviewers' suggestions on the previous version included the review by a native English speaker. However, a few typing errors have now been identified.
We hope this version is the right one.
Reviewer 3 Report (New Reviewer)
The authors did a good job in reviewing their manuscript.
Minor spelling check review.
Author Response
Thank you for your rating! It means a lot to us!
The previous suggestions of the reviewers were taken into consideration. It was also revised under the aspect of the English language by a native. At your suggestion, however, we have identified a few typing errors.
We believe that this version is the appropriate one.
best regards!
This manuscript is a resubmission of an earlier submission. The following is a list of the peer review reports and author responses from that submission.
Round 1
Reviewer 1 Report
I'm sorry to say but I have no positive opinion on the article.
Paper is generally not well written, Introduction is to long and non-focused; in Materials section there is no explanation of the statistical analyses used, Results section is extremely long and not understandable, while Discussion is shorts and poorly referenced.
Actually, I strongly believe that the paper is extracted from some longer text (thesis, disertation), which is not acceptable way of preparing the scientific papers.
Topic is generally, interesting but writing is poor and text deserves total "reset". I encourage authors to totally rewrite the paper and while doing it - ask someone experienced to help you with it. Scientific writing is specific job which can not be done intuitively.
No comments
Reviewer 2 Report
Dear Authors,
I would like to express my gratitude regarding the opportunity to review this manuscript.
At this stage the document requires improvements, below with line indication:
2-3 – Please format the title text considering the journal template and journal instructions for authors (uppercase and lowercase).
6-9 – Please format affiliations considering the journal template and journal instructions for authors. Cities are missing, also authors initials and emails.
10-19 – Please consider developing and improving the abstract, namely with content related to methodology and results.
20 – Please revise text in bold.
28 – All citations format should be revised throughout the manuscript.
24-55 – Please consider standardizing the size of the paragraphs, aiming to improve readability. This should be considered in all text, not only in these specific lines.
24-233 – Please consider reformulating the introduction section, the text is too long.
225-233 – Please consider reformulating the aim of the study and hypothesis.
236-250 – Please describe with more detail the sample and conditions. Some examples: inclusion and exclusion criteria, subjects’ routines, time of practice, swimming pool characteristics, ethical code number, written form signed by guardians? All these and other information’s should be presented to readers.
252-254 – The text does not seem justified. Please confirm.
275 - Please describe the details related to data collection (time of day, temperature, humility, places for data collection, refrain from practice? Clothes, and others). How collected the data (academic background and experience?).
234-266 – The text is too long and difficult to follow. Please consider rephrasing, presenting figure regarding procedures and other strategies aiming readers understanding.
467 – Before results, a statistical analysis subsection should be presented with all the procedures described, in detail.
467-823 – The results section is too long, impossible to be follow and understood by the readers. Reformulation is suggested, with more tables (considering the instruction for authors) and possible inclusion of figures.
824 – In the beginning of the discussion section, please consider indicating the aim of the study and the main findings, afterward discussing those with bibliographic references.
824-870 – Please consider improving the discussion section, namely developing, and improving the quality of text and content.
870 - Please indicate study limitations and suggestions for future research.
872-914 – Please consider more clear and direct conclusions, if possibly with take-home messages.
937 – All references should be carefully revised, they are not according to the journal template and journal instructions for authors.
Please carefully revise the English throughout the manuscript.
Extensive editing of English language required.
Reviewer 3 Report
The authors present an original paper investigating the relationship between psychomotor behaviors and learning 2 swimming styles (freestyle, backstroke) in children.
This is an area that has received a little attention in the literature, therefore, warrants further examination. Overall, the manuscript is written and organized to follow the logical sequence of a research purpose. Despite this strength, I have comments that need to be addressed by the authors and listed below.
ABSTRACT.
The abstract should be improved and more precise. At least, results should be presented with r and p-values.
INTRODUCTION.
The introduction is clear and present key papers related to the aim of the study. However, this section should be really synthetized and the number of references is excessive for a research paper. Indeed, this is valuable thorough the manuscript; 158 references are presented.
MATERIALS and METHODS
This section is fairly well presented but should be improved.
2.1 section. Mentioning 428 children is usefull since 76 children represent the sample of the study. Authors should clearly stipulate that consents from parents and children have been obtained. Even if the study was conducted in accordance to Oviedo convention and Helsinski declaration, authors should mentioned if their study has been approved by an Ethical Committee and provide its number.
2.3 section. All specific descriptions of used tests and specific technical details should be presented in supplementary files.
Line 363. Did you mean 360 degrees?
A section “Statistical analyses” is missing. Indeed, used analyses are integrated in the results section. It is not appropriated.
RESULTS
Results section should better be organized by using sub-sections related to each manipulated variables.
Tested hypothesis should be presented in Materials and Methods in the statistical analyses section.
DISCUSSION
First paragraph is useless, please remove it or use it elsewhere in the discussion after the second paragraph.
All the results should be better discussed. Indeed, the ratio discussion/conclusion is not acceptable.
REFERENCES
The amount of references is not appropriated for a research article.
NS